# The Direct and Indirect Influences of Adverse Childhood Experiences on Physical Health: A Cross-Sectional Study

**DOI:** 10.3390/ijerph17228507

**Published:** 2020-11-17

**Authors:** Yolanda Martín-Higarza, Yolanda Fontanil, María Dolores Méndez, Esteban Ezama

**Affiliations:** 1Institute of Legal Medicine, Government of the Principality of Asturias, 33001 Asturias, Spain; 2Department of Psychology, University of Oviedo, 33003 Oviedo, Spain; fontanil@uniovi.es; 3Mental Health Services of the Principality of Asturias, 33201 Asturias, Spain; lmmendezloli@gmail.com (M.D.M.); esteban.ezama.coto@gmail.com (E.E.)

**Keywords:** adverse childhood experiences, quality of life, physical health, social vulnerability, emotion regulation, coping strategies, attachment, social support

## Abstract

A growing and significant body of research has documented the close relationship between adverse childhood experiences (ACEs) and health outcomes in adults. Less is known about the complex pathways through which ACEs exert their influence. This article examines the direct relationship between the quality of perceived physical health and childhood adversities. The association between the adversities and the physical health with other psychological and social variables is also analyzed. Data were collected from a sample of 170 subjects, using tools to assess adverse childhood experiences, physical health-related quality of life, socioeconomic vulnerability, emotion regulation, coping strategies, attachment, and social support. Results showed a high frequency of ACEs among the adult population, and the correlation with poor physical health was highly significant. Regression equations to predict physical health also revealed the following as significant variables: wishful thinking, social withdrawal, and cognitive restructuring as coping styles; reappraisal to achieve emotion regulation; fear of rejection or abandonment, and desire for closeness in relation to attachment figures; hardship; and poor financial support. The relationship between the different forms of ACE and social vulnerability identifies the important indirect contribution of childhood adversity to health and socioeconomic conditions in adulthood.

## 1. Introduction

While the lifelong consequences of child abuse and neglect are well known [1,2,3,4], new evidence from various areas of research continues to overwhelmingly support the idea that early exposure to interpersonal violence and adverse experiences in the home has a major impact on the quality of life and health of these children as adults. The occurrence of disease, psychological dysfunction, and relationship problems as a result of violence experienced in childhood has been widely documented, which has allowed us to see how certain relational dynamics generate strategies that can compromise people’s health and well-being.

Struggling with past and present adversity can provide an environment for the emergence of adaptive but dysfunctional strategies that lead to health issues, psychological problems, or relationship difficulties. When this struggle begins in childhood, the development of personal and interpersonal resources can be hindered, leading to impaired quality of life in adulthood. The value of these findings is therefore fundamental to a better understanding of the pathways that lead to poorer health associated with early adverse experiences, and of the variables that can be included in these pathways.

### 1.1. The ACE Study and the Relevance of Childhood Adversity

The Adverse Childhood Experiences Study is one of the largest epidemiological investigations carried out to assess the relationship between various types of adverse experiences in childhood and adolescence and well-being and health in adulthood. It was the result of a collaboration between Kaiser Permanente and Centers for Disease Control and Prevention (CDC) in the United States. The study assesses the impact of childhood and adolescent maltreatment and household dysfunction on the emergence of risk factors for disease, quality of life, health system use, and mortality [5,6]. Adverse childhood experiences (ACEs) are defined as experiences of maltreatment or adversity that occur in the first 18 years of a person’s life. They are measured through different categories of experiences. One of the most common categorizations comprises 10 types of adversity that produce the “ACE score”: three types are abuse (physical, emotional, and sexual); two are neglect (physical and emotional); and five are categorized as household dysfunction.

In assessing the relationship between childhood exposure to adversity and health risk behaviors and different diseases, results demonstrated a strong and graded relationship between the number of categories of exposure to maltreatment and each of the risk factors studied. For example, an experience of four or more categories of exposure was associated with a four- to 12-fold increased risk for alcoholism, drug misuse, and suicide. A significant relationship was also found between the number of categories of adverse experiences and the following medical problems: ischemic heart disease, cancer, chronic bronchitis or chronic obstructive pulmonary disease (COPD), hepatitis or jaundice, and skeletal fractures [5].

Following on from this pioneering investigation, numerous other studies have gone on to confirm this relationship, in identifying the direct effects on health [7,8,9,10,11] and the health risk behaviors [12], or in investigating the variables that mediate between adverse experiences and health outcomes [13,14,15,16,17].

Given the strong association between ACEs and health, there is a need for a better understanding of the mechanisms involved. This knowledge would serve to improve prevention and intervention strategies that address the underlying factors, and would ultimately influence the main causes of morbidity and mortality. This analysis is particularly relevant given that one of the variables strongly related to this association is the experience of abuse, neglect, and other childhood adversities related to the actions of other people.

### 1.2. The Impact of Adversity on Quality of Life

One of the aims of this study was to examine the relationship between childhood experiences and physical health-related quality of life domain (PH-QOL) in adults.

The World Health Organization (WHO) defines quality of life as “an individual’s perception of their position in life within the context of the culture and value systems in which they live and in relation to their goals, expectations, standards and concerns [18].” This and other definitions highlight the subjective and multidimensional nature of the concept, incorporating a wide range of aspects of a person, such as physical health, psychological status, degree of independence, social relationships, environmental characteristics, or personal beliefs [19,20].

From this perspective, an acceptable health status is considered a positive factor for quality of life. Illness, in contrast, poses a threat to well-being and implies the need to adapt to changes and possible losses (of quality of life, activities, relationships, or life itself) as well as changes in the evolutionary tasks people face at each life stage. Thus, physical health is a further indicator of quality of life in which objective and subjective elements can be distinguished. Personal perceptions about health have an influence on conceptions of subjective well-being. Such is the link between the two concepts that health-related quality of life (HRQL) as a measure of health status is in some cases considered to be synonymous with overall quality of life, and in others to be a component part of it. The relationship between impaired HRQL and childhood adversity has been studied extensively [21,22,23], although other research also indicates that not all individuals exposed to childhood adversity exhibit somatic symptoms in adulthood [24,25]. Since it is rare that a direct and unequivocal relationship between two variables is established, such diversity in results is to be expected. To progress in our knowledge requires investigation into the role of other life circumstances that may affect the results. Our study assesses the direct relations of ACEs with PH-QOL as well as possible indirect associations through other psychological and social processes

### 1.3. The Role of Childhood Adversity in Adult Socioeconomic Vulnerability

Part of our study sample consisted of people in positions of social vulnerability. This concept refers to positions of weakness, job insecurity, and fragility in relational ties. These conditions affect—to varying degrees—a range of social groups, not just those living in circumstances usually associated with poverty [26]. Many individuals in today’s society, not only those with limited economic resources, may find themselves in this social condition of risk. This vulnerability puts people in a situation of present adversity, which in turn can multiply the impact of past adversity, generating contexts in which health and quality of life are seriously impaired.

Childhood adversity, social vulnerability, and harm to mental and physical health and quality of life are all closely related. This link has been explained through various pathways, and there has even been some discussion as to whether unfavorable socioeconomic conditions should be considered as another type of adverse experience or as an independent factor contributing to adult outcomes [27,28].

While the impact of ACEs is considered an important issue, few studies have addressed it in a socially vulnerable population, which has been considered in this study. The relationship between childhood adversity and socioeconomic situation is far from simple. On the one hand, several studies have shown the relationship between ACEs and socioeconomic difficulties in adulthood, in many cases independently of or partially dependent on demographic factors or the socioeconomic situation of the family of origin [1,29,30,31,32]; on the other hand, results demonstrate the impact of ACEs decades later, even without social disadvantage or contributing sociodemographic factors [27]. The relationships between these variables remain unclear.

However, the hypothesis remains that the relationship between ACEs and health is connected with socioeconomic factors that in many cases generate problems in other areas (e.g., school performance, social functioning). This situation produces “chains of risk” and adversities in all aspects of an individual’s life, by weakening their capacity to access higher income, obtain social support, get professional help, maintain healthy habits, or avoid exposure to other adversities—precisely what defines socioeconomic vulnerability—and results in a deterioration of health [29,31,32,33,34].

Despite the need for further research into the factors involved in this relationship, it is clear that wherever they occur in the succession of difficulties generated by any form of abuse, socioeconomic factors play an important role in the transmission of the effects of maltreatment, with the consequent decline in the quality of life and health of the people involved.

### 1.4. Coping and Emotion Regulation

To explain this process whereby past adversities carry over into current health status, some studies have focused their attention on strategies that people employ to cope with situations of distress [35].

Coping as a concept encompasses the range of strategies used in the face of stressful life events in order to adjust to the new situation. It has been closely linked to health because the way in which a person tackles threatening situations can greatly affect their physical and psychological state. Physiological activity is in fact a tool that people use to respond to their environment. The coping strategies employed to deal with adversity can lead to better or worse prospects, in terms of the cost of putting them into operation as well as their potential to succeed in overcoming the problem. Different reactions generate diversity in people’s capacity to adapt to adversity; they also explain, at least in part, many of the physical and mental health outcomes. Most studies conducted to date have linked different coping strategies to the presence of health problems. Historically, strategies that facilitate problem solving (e.g., task-focused) are viewed as adaptive, whereas those that encourage focus on emotions are not [36]. While the role of coping styles seems proven [37], the results obtained in the literature do not provide a clear pattern of the weight of the different coping strategies, which fluctuates depending on the subject matter [38,39,40,41].

There has been a limited amount of research into the link between ACEs and coping style. A history of maltreatment has been associated with increased avoidance and emotion-focused coping, and with decreased task-focused coping [42,43,44,45]. Over the last five years, there has been a rise in studies establishing a relationship between childhood adversity and emotion regulation difficulties [46,47]. Schimmenti and Caretti’s (2018) starting position was that psychological trauma affects the capacity to understand and process emotions, especially when the harm is inflicted at an early age and by significant people such as attachment figures [48]. They reviewed the literature and confirmed the view that child maltreatment is a potential precursor to problems in identifying and describing one’s own emotions. Models have also been developed suggesting that emotion regulation mediates the relationship between childhood adversity and adult health (physical health β = 0.07, *p* = 0.002) [49], although the relationships were always lower when compared with psychological health.

The impact of adversities on organic disease is therefore evident, which is why in our analysis we consider physical health as being linked to—and not detached from—the goals that people set for their adaptation and development; and to the strategies they use to achieve these goals, involving changes in physiological processes.

### 1.5. The Influence of Relationships: Attachment Style and Social Support

The relationship between adversity and emotion regulation is also relevant when considering the formation of attachment bonds, a subject that has attracted a considerable amount of research attention in recent decades. It is based on the idea that emotion regulation is organized via the baby’s dyadic relationship with its caregivers during the different phases of attachment development. Through the child’s attachment behaviors and adult-caregiver actions, including their emotional availability and the quality of communication of their emotions, mechanisms are established to regulate emotional arousal and behavioral expression of emotions [50,51,52,53]. Babies who have taken part in relationships of effective dyadic regulation—in which significant others have been available and have responded to their arousal—do not feel disorganized by triggers and quickly regain equilibrium. This dynamic generates a secure pattern of attachment associated with a willingness to participate in emotionally stimulating situations, with an acceptance to express emotions and know that these are not by themselves threatening, and with an expectation that others will interpret this expression as an acceptable message. This secure working model is characteristic of children who are highly curious, have a taste for exploration and affective expression, especially in social situations, and who effectively turn to others when their own capacities may be insufficient [52,53]. However, affectional bonds can also be insecure—and this is more likely to be the case where there is a history of ACEs—which can give rise to difficulties regulating emotions and forming social relationships when faced with threatening or stressful life situations.

Adult attachment styles are characteristic patterns of response to distress [54]. People who experience limited discomfort with closeness (avoidance) and limited fear of rejection or abandonment (anxiety) show greater control of negative feelings, recognize distress, and seek support and comfort in others. The role attributed to others as a possible source of support in the regulation of distress is manifested in interaction strategies that are deployed either to avoid closeness or to seek it out intensely. These strategies are learned in infancy, determined by the availability (or lack thereof) of the primary caregivers and how receptive they were to the child’s search for closeness; they will indirectly impact the type of reactions and relationships that an infant who has experienced abuse will have as an adult. Studies that have explored the relationship between affectional bonds, adversity, and health—often separately—highlight insecure attachment as a factor that needs to be taken into account [55,56,57,58,59,60,61,62,63].

There is an undeniable link between close and caring relationships and health and well-being at all life stages. That said, little is known about the specific pathways through which close relationships promote optimal well-being. Some attempts at explanation have shown how certain interpersonal processes, such as attachment and social support, improve the ability to cope with adversity and promote a more active attitude toward life. Social support is therefore considered a protective factor for health in individuals experiencing stress. Social support generally refers to actions carried out on behalf of an individual by others, such as friends, family, and coworkers. These people can provide various types of support, including instrumental, informational, and emotional support. Among the various models used to explain the influence of social support on health, one of the most relevant today is based on the concept of stress buffering [64]. We should also remember that social support can exert a direct effect on health, that is, the support itself is health promoting (e.g., financial support can facilitate health care or quality of life). Therefore, the influence of social support as a protector against stress and a promoter of health must be understood in the context of the complex interrelationships that exist between the demands for support and social networks and structures. Feeney and Collins proposed two social support functions that help people to thrive in life situations, even in adversity: source of strength support (support to reduce the impact of stress and to help overcome it) and relational catalyst support (support to seize opportunities for improvement). These functions are mediators through which adversity is likely to lead to a better life in the long term [65,66]. Social support is therefore another important factor that is closely related to quality of life, to the extent that it may even be considered another dimension of the concept, in the same way as health or material conditions [20,36,67].

Our study will explore the association of PH-QOL and ACEs with attachment relationships dominated by a fear of rejection and abandonment, by a preference for independence, or by a desire for closeness to attachment figures in stressful or difficult situations. This examination will also take into account the support received from family and caregivers in childhood or from close others in adulthood.

### 1.6. Aims

The overall aims of the study were to test the hypothesis associating the presence of ACEs with poorer PH-QOL, and to explore the role of other variables in this relationship.

Initially, we sought to verify whether the direct relationship between adversity and health outcomes—the subject of much research over the last three decades—was also found in our study sample. To this end, we assessed the direct influence of childhood adversity on self-reported physical health, in terms of both the number and types of ACEs. The purpose of this first analyses was to determine whether the accumulation of abuse, neglect, or difficult interpersonal experiences in childhood and/or their specificity impact on the long-term health of the person who experienced them.

We also evaluated the relationship of other variables to PH-QOL. These studies included age and gender; social vulnerability factors; relational variables such as attachment style or social support; coping strategies for emotion regulation; and other behavior styles in response to stress. The purpose of these second analyses was to identify other predictor variables for PH-QOL.

Given the difficulty of establishing direct relationships, the last aim was to analyze the indirect relationships through which ACEs are associated with these predictor variables. This step provided insight into the pathways through which the harm caused by adversity could spread to other areas of a person’s functioning and adversely affect their health. We performed this analysis for each of the ACEs in order to achieve a clearer understanding of their differential impact based on the contribution of each of the predictor variables.

## 2. Materials and Methods

### 2.1. Sample and Procedure

The sample was made up of two groups of people over the age of 18: socially vulnerable people who were receiving support from Social Services and other welfare resources of the Principality of Asturias (Spain); and people who did not indicate they had basic needs. The conditions of participation were the same for both groups. All subjects were adults and were not affected by intellectual disability, organic brain damage, central nervous system problems, or communication difficulties that might impede evaluation. The study was approved by the Research Ethics Committee of the University of Oviedo, and the process was carried out in compliance with the ethical principles of the Helsinki Declaration. Those who met the requirements were informed of the aims of the research. After agreeing to participate in the study they were asked to sign an informed consent form. The background characteristics of the sample are shown in Table 1.

Our sample size was 170 subjects, of whom 118 (69.4%) were women. In all, 64.8% of the sample was aged between 30 and 49, 18.2% between 50 and 69, and 15.8% between 18 and 29. As regards marital status, 31.2% of the participants were married, 22.9% were separated or divorced, and 19.4% were single. In terms of educational attainment, the most represented levels were university (35.5%), followed by primary education (22.5%). The most common employment statuses cited were employed (42.4%) and unemployed (35.9%). For most participants, the main source of income came from their own job or a family member’s job (58.8%), and for 23.5% the main source of income was government benefits.

### 2.2. Measures

(a)The quality of life related with physical health was assessed using the physical health domain (HRQoL_physical_) of the Spanish adaptation of the short form of the World Health Quality of Life Questionnaire-BREF (WHOQOL-BREF) developed by the WHOQOL Group [68]. This domain contains seven averaged items with five response options (e.g., from 1 = very dissatisfied to 5 = very satisfied), and in our sample yielded a reliability of α = 0.84. The correlation between (HRQoL_physical_) and the number of chronic health problems was rho = −0.536 (*p* < 0.000). This result supports the validity of the health questionnaire used in the analyses.(b)A Spanish translation of the Adverse Childhood Experiences (ACEs) Questionnaire (Felitti et al., 1998) [5] produced by the authors of this study was used to evaluate adverse experiences of abuse, neglect, and household dysfunction during the first 18 years of life. Ten adverse experiences were assessed: physical, emotional, and sexual abuse; physical and emotional neglect; parental divorce or death; witnessing domestic violence; household substance abuse; household mental illness or having a family member attempt or die by suicide; and having an incarcerated household member. The frequency with which the person experienced them was determined by summing the 10 items. In our sample, the reliability of the questionnaire was α = 0.79.(c)Socioeconomic vulnerability was assessed via a set of items constructed for this study, using the hardship scale, which measures difficulty in covering unavoidable costs, in accessing training or employment, and in carrying out administrative procedures in the previous three months (0 = no hardship, 1 = once, and 2 = more than once) (α = 0.65); and also using the variable government benefits, which covers support received to resolve financial, social, environmental or housing, family, and personal difficulties (from 0 = no benefits to 5 = benefits in all areas).(d)The Coping Strategies Inventory (CSI) by Tobin, Holroyd, Reynolds, and Kigal (1989) in its Spanish version (Inventario de Estrategias de Afrontamiento), by Cano, Rodríguez, García (2007) [69], evaluated eight coping strategies on a five-point Likert scale. Respondents had to briefly describe a stressful event or situation that had happened to them in the previous month and answer 40 statements, on a scale of 0 (not at all) to 4 (completely), rating the extent to which they experienced certain thoughts or feelings or displayed certain behaviors in relation to the chosen situation. Scores were calculated by summing the responses in each scale. The reliability for this study was problem solving (α = 0.89 in our sample), cognitive restructuring (α = 0.83), social support (α = 0.87), express emotions (α = 0.83); problem avoidance (α = 0.66), wishful thinking (α = 0.77), social withdrawal (α = 0.80), and self-criticism (α = 0.91).(e)A Spanish adaptation of the Emotion Regulation Questionnaire (ERQ; Gross and John, 2003) by Cabello et al. (2013) [70] was used to assess emotion regulation. This questionnaire measures two regulation strategies: the tendency to regulate emotions by cognitive reappraisal (α = 0.67) or by expressive suppression (α = 0.77). The items are scored by summing the responses on a scale of 1 (strongly disagree) to 7 (strongly agree).(f)The Scale of Preferences and Expectations in Close Interpersonal Relationships (EPERIC) (Fontanil, Ezama, and Alonso, 2013) [71] assesses adult attachment styles. The questionnaire is made up of 22 items, each with a Likert-type response on a scale of 1 to 5, where 1 means “is nothing like what happens to me” and 5 means “is very much like what happens to me”. It assesses fear of rejection or abandonment (FRA) (α = 0.87), desire for closeness (DC) (α = 0.71), and preference for independence (PI) (α = 0.75).(g)The Cuestionario Breve de Apoyo Social (CBAS) (Sandín, 2006) [72]. This is a series of 12 questions evaluating the social support that a person receives. This research differentiated between support provided by people with whom the person lived before adulthood, such as family or other caregivers (referred to in this paper as the childhood social network, or CSN); and other close people (partners, friends, etc.) with whom they did not live before the age of 18 (labeled the adulthood social network, or ASN). This support was rated between 0 (never) and 4 (almost always). Scores were obtained for financial support (CSN: α = 0.87; ASN: α = 0.84), emotional support (CSN: α = 0.94; ASN: α = 0.88), support through appreciation (CSN: α = 0.93; ASN: α = 0.89), and support through advice (CSN: α = 0.93; ASN: α = 0.87).(h)Number of chronic health problems. To determine the presence of long-term illness and health problems, we referred to the list of 29 chronic medical conditions used by Min et al. (2013) [15] in their study on the association between child maltreatment and poor physical health and the presence of organic diseases in adulthood. Participants answered yes or no for each health problem. This information was used to create a variable by adding the number of items that had received a yes response.

### 2.3. Data Analysis

The supposition of normality was checked using the Kolgomorov–Smirnov test, finding non-normality of distribution for almost all the variables. Following that, in order to achieve the first aim, Spearman’s correlation tests were performed to verify whether the direct relationship between adversity and health outcomes was also found in our study sample, and to compare the sample for each of the ACEs, the Mann–Whitney U test was used.

To reach the second aim, Spearman’s correlation and five multiple linear regression analyses following the stepwise method (forward selection) were performed using the physical health domain as a dependent variable. Possible predictors considered were age, gender, number of ACEs, hardship and government benefits, coping styles, emotional regulation, attachment, and social support. These variables were analyzed in four separate blocks. The stepwise method (forward selection) was chosen because it continuously checks the contribution of each independent variable as it is added to the regression model, eliminating any variables whose contribution to the model is better explained by another. In each case, the supposition of independence of residuals was verified with the Durbin–Watson statistic, and the normality of their distribution, which justified the use of the analysis despite the non-normality of the variables, through the Kolgomorov–Smirnov test.

In order to meet the third aim, the scores obtained in each of the predictor variables were then compared using the Mann–Whitney U statistic test in order to examine another pathway through which ACEs can influence PH-QOL.

The bilateral level of significance established prior to all the tests was 0.05. The statistical analyses were carried out using SPSS version 20.0 (IBM SPSS, Inc., Chicago, IL, USA).

## 3. Results

### 3.1. Descriptive Statistics

#### Descriptive Analysis of HRQoL_phisical_, ACE, and Socioeconomic Vulnerability Data

The mean for physical health was 3.4 (min. 1.14, max. 4.86), the median 3.5, and the standard deviation 0.83. As a score closer to 5 suggests a better perception of health, our results indicate a normal to fairly good perception of health, along with some dispersion of situations perceived as fairly bad to almost optimal.

When asked about long-term illness, 63.7% of the sample indicated they had at least one chronic condition. The median was 2, the mean 2.36, and the standard deviation 1.68. The most frequently cited conditions were nonspecific gastrointestinal problems (17.6%), asthma and tachycardia (15.5%), gastritis or colitis (14.7%), hypercholesterolemia or hypertriglyceridemia (14.3%), hypertension (11.2%), irritable bowel syndrome (10%), and fibromyalgia (9.4%).

With regard to ACEs, the mean for adverse childhood experiences was 3.33 and the standard deviation 2.66 and only 16% of the sample subjects reported no adversity, whereas 43.2% had experienced four or more types, including four participants who had experienced all ACEs. This is therefore a sample with a high incidence of ACEs. Figures on the frequency and percentage are shown in Table 2. There are noticeable differences between our sample data and those of the original ACE Study, in which the percentage of people who reported no adversity (ACE = 0) was 36%, and 16% had a score of 4 or more points.

The most frequent forms of adversity experienced by our sample were emotional abuse and neglect, while the least represented were physical neglect and having an incarcerated household member (11.8%) (see Table 3).

Once again, there were differences when we compared our results with the baseline study. In the original ACE Study [5], the most frequently cited forms of adversity were physical abuse and sexual abuse. There was some agreement, however, in the least frequent category, which was the incarceration of a household member.

For the results on social vulnerability, the hardship variable had a mean of 0.42 and a standard deviation of 0.57. We are reminded that a score above zero implies difficulties in meeting basic needs, in accessing training or employment, or in carrying out administrative procedures in the previous three months. As for government benefits, 47.9% were not in receipt of any type of benefit, 30.8% were receiving one type of benefit, 15.4% were receiving two, and the remaining 5.9% between three and four.

### 3.2. Association Analysis

Aim 1: The correlation between (HRQoL_physical_) and the number of ACEs was significant (rho = 0.188, *p* < 0.005). The analysis by type of ACE revealed that emotional neglect (U = 2370.50, *p* = 0.000), physical neglect (U = 977.00, *p* < 0.05), and having a parent with a mental illness or who has attempted or died by suicide (U = 2695.00, *p* < 0.05) were associated with poorer PH-QOL.

Aim 2: This direct relationship between ACEs and poorer quality of life can be supplemented by an analysis of other variables that may be related to both. The first step was to examine the correlations between (HRQoL_physical_) and the rest of the predictor variables, and then to perform regression equations to show the predictive weight of these variables (Table 4).

### 3.3. Regression Analysis on Significant Variables for Physical Health Quality of Life

First, we performed forward stepwise multiple regression analysis on the statistically significant variables (see Table 4). Our criterion was HRQoL_physical_, and our predictors were four blocks of variables: age and number of ACE types (block 1); hardship and government benefits (block 2); coping strategies and emotion regulation (block 3); and social support and attachment style (block 4).

With the predictor variables of the four block equations, we performed a final regression (see left-hand side of Table 5). The regression equations to predict physical health quality of life gave the following results:

For block 1, the regression equation gave two variables that explained 11.8% of the variance (R = 0.344, R^2^ = 0.118): age (β = −0.274, *p* = 0.000) and number of ACE types (β = −0.213, *p* = 0.005).

Block 2 reached a model that introduced one variable—hardship—explaining 4.7% of the variance (R = 0.218, R^2^ = 0.047, β = −0.322, *p* < 0.005).

The block 3 model introduced four variables that explained 21.9% of the variance (R = 0.468, R^2^ = 0.219): using wishful thinking (β = −0.248, *p* = 0.001), social withdrawal (β = −0.223, *p* < 0.005) and cognitive restructuring (β = 0.171, *p* < 0.05) as coping styles; and reappraisal to achieve emotion regulation (β = 0.177, *p* < 0.05).

From block 4, the regression equation included three variables that explained 17.5% of the variance (R = 0.418, R^2^ = 0.175): financial support from ASN (β = 0.242, *p* <0.005), fear of rejection or abandonment (β = −0.244, *p* < 0.005), and desire for closeness in relation to attachment figures (β = 0.153, *p* < 0.05).

Aim 3: In order to examine another pathway through which ACEs could be related to PH-QOL, the scores obtained in each of the predictor variables were then compared using the Mann–Whitney U statistic, based on respondents’ answers on whether they had experienced a particular ACE or not.

The right-hand side of Table 5 shows the statistically significant relationships for the predictor variables associated with HRQoL_physical_. The analysis revealed that hardship in the ability to meet basic needs was associated with nine of the 10 ACE types. The only ACE that did not affect the degree of hardship was parental incarceration. At the other extreme, the scores pertaining to desire for closeness did not vary significantly based on any of the 10 ACEs. The distribution of fear of rejection or abandonment was statistically different for emotional, physical, and sexual abuse, emotional neglect, problems with drugs, and parental mental illness or suicide. The same was also true for financial support from respondents’ ASN, which was associated with emotional and physical abuse, emotional and physical neglect, problems with drugs, and household mental illness or having a parent attempt or die by suicide.

For the coping and emotion regulation strategies associated with physical health as per the regression equations, the U results on the right-hand side of Table 5 show that the use of wishful thinking also appears to be associated with experiences of physical and sexual abuse, while social withdrawal as a strategy to cope with problems was more common in people who reported having experienced emotional neglect. The strategy of cognitive restructuring did not vary according to adverse experiences. Scores for reappraisal (emotion regulation) varied depending on the presence of two experiences: alcohol or drug problems and parental incarceration.

## 4. Discussion

Our study into the impact of ACEs on physical health is one of a few not only to have been conducted in Spain but also to have considered the role of mediating variables, including socioeconomic variables. Our results complement those of the first ACE Study, in that they revealed a considerable prevalence of ACEs among a population that was middle class, mostly white, educated, and with access to health insurance. Other studies have supported the idea that the consequences of ACEs can affect any population, not only those usually associated with violence and maltreatment. In our study—comprising a heterogeneous sample, albeit with many participants recruited via mental health and social services consultations—the frequency of ACEs was even higher when compared with the study by Felitti et al. and subsequent research carried out in Europe and North America [73]. Almost half of our sample had experienced four or more types of ACEs.

With this sample, the relationship between ACEs and perceived physical health is clear. The correlation between the number of ACEs and PH-QOL is significant. The greater the number of ACEs, the stronger the relationship with poorer self-reported PH-QOL. On examining the importance of each individual ACE, we found that emotional neglect, physical neglect, and living with someone who had a mental illness and/or who attempted suicide appear to be linked to poorer PH-QOL. In this regard, it is worth noting that it was not physical abuse that was directly linked to health-related problems for our sample but rather the imbalance between needs and care, in line with previous research [10]. These results highlight the need to pay particular attention to neglect in child maltreatment prevention and intervention programs, because the identification of a mental disorder in a family could be indicative of family environments where neglect is present to the same or to a greater extent than physical violence. A closer examination of this relationship may require more detailed information about relational difficulties involved in the experience of household mental illness and suicide (whether attempted or completed).

By extending our analysis to investigate the influence of other variables, we have seen how adverse experiences impact both directly and indirectly on PH-QOL. Now the number of ACE types is not an irreplaceable variable and other variables are better predictors of PH-QOL. In addition to age (the variable with the greatest weight in the regression equation), the relationship between physical health and the following variables is significant (in descending order of influence): social withdrawal, cognitive restructuring, wishful thinking, hardship, and financial support from ASN; although, with the exception of cognitive restructuring, the value of these variables changes depending on the ACE types, such that the influence can be said to be indirect.

Multiple theories have been advanced to explain the relationship between ACEs and health [13,14,15,16,17,31,34]. Among the most accepted are those that link adverse experiences to the development of risk behaviors and those that refer to the effects of cumulative stress on neurodevelopment. Other models have focused on the strategies that people put in place to cope with adversity, their relational preferences, or how they rely on the support of others. Our view is that people deal with ACEs by using the knowledge, skills, and preferences they have already developed or are developing. They also make use of the scaffolding that comes from the actions of their social network, especially those of their attachment figures. The expectations and the procedural and declarative knowledge developed to adapt to childhood and adolescent adversity influence a person’s later attempts to overcome difficulties and seize opportunities for adaptation in response to adverse situations that pose a threat to quality of life [66]. However, the use and construction of such skills and knowledge comes at an organic, physiological and anatomical, cost that increases the likelihood of ill-health [74]. Thus, to the risk of being harmed by the consequences of behaviors, we must add the cost of adaptation: the allostatic load. [14,31].

What is the role of socioeconomic conditions in this model? The block regression equations showed that the four variables relating to emotion regulation and coping style (wishful thinking, social withdrawal, cognitive restructuring, and reappraisal as an emotion regulation strategy, in descending order of importance) best explained the variation in quality of physical health (22% of the variance). The three variables for attachment and social support (fear of rejection or abandonment, financial support from ASN, and desire for closeness to attachment figures) explained 17.5%, while in the block for social vulnerability, the only predictor was hardship, accounting for just 5% of the variance. That being said, the inclusion of this latter variable in the equation suggests that the information it provides is not covered by the other variables. Poverty and vulnerability as a factor have been closely related to abuse and neglect, to the extent that some researchers consider them to be another ACE [75]. In our case, we did not assess this factor as an early adversity but as one that is present to varying degrees in adulthood. More than half of the subjects in our sample were in receipt of some form of government benefit to meet financial, social, family, personal, or housing needs; the level of hardship was also considerable. In our study, hardship was related to nine of the 10 ACEs. In other words, the level of socioeconomic vulnerability was significantly higher in the subjects who had been exposed to almost every adverse experience when compared with those who had not. It was a similar story for financial support from ASN. In this case, six of the 10 ACE types had a significant influence. A lack of socioeconomic resources, therefore, seems to play an important role in the quality of health, even in a country where health care is free and universal. The question is, is hardship a consequence of strategies to adapt to early adversity, or is it rather something that accompanies or promotes adversity? It is not possible to fully answer this question from the information we requested of participants. Nonetheless, the social support variable that correlates significantly with physical health is not financial support from the family of origin (which makes up the bulk of a person’s social network when they are under 18) but rather financial support from their support network in adulthood, which leads us to believe that current socioeconomic resources can indeed be conditioned by adaptations to early adversity. Our assumption is confirmed by the results of other work showing that adults who were exposed to ACEs experienced worse socioeconomic conditions, regardless of their starting conditions in childhood [1,30,31,32,33,34]. In general, we would agree with fellow researchers who refer to ACEs as links in “chains of risk” caused by the accumulation of adversities that affect people’s ability to successfully carry out life tasks [31,32].

The number and circumstances of the participants do not allow our results to be generalized. Information provided by retrospective reports is in no way incontestable, although there is evidence to suggest that it can be sufficiently reliable [76,77,78]. Our results can serve to elicit some hypotheses for further exploration with larger and more heterogeneous samples. What we have found reinforces the view that improving the quality of health for the adult population of tomorrow involves reducing the adverse experiences for the children and adolescents of today. Such a change will not come about unless adult caregivers are provided with better parenting strategies, better living conditions, and better knowledge on how to give and receive care. New forms of intervention to prevent child abuse and neglect are required. Equally, there is a need to acknowledge that time is often not a healer for adults who as children experienced abuse, neglect, and family conflict. Efforts to adapt to adversity are not always successful and often make it more difficult to adapt to future adversities and opportunities. For that reason, studying the relationship between psychological dysfunction and adverse experiences in a similar sample seems to be a logical follow-on to this research. We believe that a lack of knowledge is one of the reasons why we health professionals do not ask about early adversity and the impact it has on our adult clients’ daily lives, and the neglect and maltreatment can end up being reproduced in professional settings.

## 5. Conclusions

The data from our study converge with other research in highlighting the relative frequency of ACEs in adult populations from different backgrounds. Similarly, our study also pointed to the direct relationship between these childhood experiences and poorer physical health, with neglect or the imbalance between needs and care contributing as a major factor to this association. Other factors play a role, as revealed in our analysis of the variables with a direct association. The most important of these were age, strategies people use to cope with present adversity (social withdrawal, cognitive restructuring, and wishful thinking), and the quality of relationships (financial support from the person’s social network). Next came hardship, which in addition occurred with most of the forms of adverse experience. This finding further acknowledges the contribution of socioeconomic conditions to health and well-being, and supports the need to strengthen health-promoting factors that include social support to buffer the effects of early adversity and social disadvantage.

Children who have been maltreated suffer important and lasting consequences that are strongly linked to the absence of adequate care throughout their lives. These consequences persist into adulthood through the use of more dysfunctional coping and relationship strategies, and are reflected in socioeconomic status as well as a worse self-perceived quality of life. In short, abuse and neglect can exacerbate social differences and generate worse health expectations for people who experience them.

## Figures and Tables

**Table 1 ijerph-17-08507-t001:** Description of the study sample. Sociodemographic variables.

Variable	Sociodemographic Variables	*N*	Percent%
Gender	Female	118	69.4
	Male	52	30.6
Age	18–29 years	27	15.8
	30–49 years	110	64.8
	50–69 years	31	18.2
	Over 70 years	2	1.2
Education level	No education or primary education unfinished	3	1.8
	Primary education	38	22.5
	Secondary education	17	10.1
	High school diploma/Basic vocational training	28	16.6
	Higher-level vocational training	23	13.6
	University	60	35.5
Employment status	Self-employed	13	7.6
	Employee	72	42.4
	Unemployed	61	35.9
	Retired	6	3.5
	Disabled	9	5.3
	Student	9	5.3
Main source of income	Job	100	58.8
	Social Security	12	7.1
	Government benefits	40	23.5
	Other	18	10.6

**Table 2 ijerph-17-08507-t002:** Frequency and percentage of adverse childhood experiences (ACEs).

Nº ACEs	Frequency	Percent %
None	27	16.0
One	29	17.2
Two	17	10.1
Three	23	13.6
Four	21	12.4
Five	11	6.5
Six	17	10.1
Seven	11	6.5
Eight	7	4.1
Nine	2	1.2
Ten	4	2.4

**Table 3 ijerph-17-08507-t003:** Frequency and percentage of ACEs by type.

ACE Type	Frequency (*N* = 169)	Percent %
Emotional abuse	83	49.1
Physical abuse	67	39.6
Sexual abuse	38	22.5
Emotional neglect	80	47.3
Physical neglect	20	11.8
Parental divorce or death	67	39.6
Witnessing domestic violence	44	26.0
Household substance abuse	74	43.8
Mental illness or family member attempt or die by suicide	71	42.0
Incarcerated household member	20	11.8

**Table 4 ijerph-17-08507-t004:** Spearman correlations between predictor variables and physical health.

Spearman’s Rho	Physical Health
Age	−0.239 **
Gender (male = −1, female = +1)	−0.132
Number of ACE types	−0.198 **
Hardship	−0.226 **
Number of types of government benefits	−0.142
Reappraisal to achieve emotion regulation	0.192 **
Suppression to achieve emotion regulation	−0.205 **
Problem solving as a coping style	0.154
Self-criticism as a coping style	−0.150
Emotional expression as a coping style	0.063
Wishful thinking as a coping style	−0.326
Seeking social support as a coping style	0.145
Cognitive restructuring as a coping style	0.221 **
Problem avoidance as a coping style	−0.046
Social withdrawal as a coping style	−0.293 **
Financial support from CSN	0.114
Emotional support from CSN	0.138
Support through appreciation from CSN	0.193 **
Support through advice from CSN	0.098
Financial support from ASN	0.290 **
Emotional support from ASN	0.264 **
Support through appreciation from ASN	0.316 **
Support through advice from ASN	0.261 **
Fear of rejection or abandonment by attachment figures (FRA)	−0.327 **
Desire for closeness to attachment figures (DC)	0.157 **
Preference for independence from attachment figures (PI)	−0.083

Note: ** *p* < 0.01. CSN: childhood social network; ASN: adulthood social network; FRA: fear of rejection or abandonment; DC: desire for closeness; PI: preference for independence.

**Table 5 ijerph-17-08507-t005:** Influence of physical health domain (HRQoL_physical_) on predictor variables.

Block	Predictor Variable	Beta in Block Equation	R^2^ of Block Equation	Beta in General Equation	R^2^ of General Equation	Adverse Experiences Influencing Predictor Variables				
						Varia-ble	Mid-Range of NO	Mid-Range of YES	Mann-Whitney U	Sig.
1	Age	−0.274	0.118	−0.266	0.323					
	No. of ACEs	−0.213								
2	Hardship	−0.218	0.047	−0.151		EA-1	71.25	98.07	2401.5	0.000
						PA-2	73.39	101.25	2261.5	0.000
						SA-3	79.47	101.71	1816	0.007
						EN-4	77.13	92.61	2871.5	0.025
						PN-5	79.94	118.23	805.5	0.000
						PD/D-6	75.66	98.16	2464.5	0.001
						WDV-7	75.91	108.72	1662.5	0.000
						SA-8	73.08	99.36	2383.0	0.000
						MI/S-9	72.23	101.26	2253.5	0.000
3	Wishful thinking as CS	−0.248	0.219	−0.185		PA-2	72.25	87.73	2348.0	0.035
						SA-3	72.93	97.07	1491.5	0.005
	Reappraisal to achieve emotion regulation	0.177				SA-8	91.68	75.38	2803.0	0.031
						IN-10	87.73	60.60	1002.0	0.019
	Social withdrawal (CS)	−0.223		−0.220		EN-4	66.35	91.97	2037.5	0.000
	Cognitive restructuring (CS)	0.171		0.195						
4	Financial support from ASN	0.242	0.175	0.140		EA-1	91.76	55.44	2775.5	0.029
						PA-2	90.44	73.24	2629.0	0.022
						EN-4	91.42	74.78	2747.5	0.024
						PN-5	87.25	56.15	913.0	0.006
						WDV-7	91.16	62.27	1750.0	0.001
						SA-8	91.17	73.97	2698.5	0.020
	Fear of rejection or abandonment by attachment figures	−0.244				EA-1	72.02	98.45	2452.5	0.000
						PA-2	75.97	98.75	2496.0	0.003
						SA-3	80.94	99.00	1957.0	0.045
						EN-4	69.69	102.04	2197.0	0.000
						SA-8	76.56	95.83	2713.5	0.011
						MI/S-9	77.53	95.32	2746.5	0.020
	Desire for closeness to attachment figures	0.153								

Note: CS = coping style; EA-1 = emotional abuse; PA-2 = physical abuse; SA-3 = sexual abuse; EN-4 = emotional neglect; PN-5 = physical neglect; PD/D-6 = parental divorce or dead; WDV-7 = witnessing domestic violence; SA-8 = household substance abuse; MI/S-9 = household mental illness or suicide; IN-10 = parental incarceration.

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
