# Peer review of "The Direct and Indirect Influences of Adverse Childhood Experiences on Physical Health: A Cross-Sectional Study"

_ijerph, 2020, doi:10.3390/ijerph17228507_

Round 1

Reviewer 1 Report

The conceptual base for this manuscript is well-supported by significant research and relevant literature. The research design is appropriate and applied with fidelity.

Results are presented with clarity in the narrative, although the format of some of the tables makes a bit of difficult reading.

The discussion of results is linear and enriched with relevant studies.

Line 14-16  Consider separating Line 15 and Line 16 as separate points of study.

Line 17  Clarify “using tools to access ACEs.” Characterize the tools.

Author Response

Point 1. Line 14-16  Consider separating Line 15 and Line 16 as separate points of study.

Response 1. Lines 16-18.  This article examines the direct relationship between the quality of perceived physical health and childhood adversities. The association between this adversities and the physical health with other psychological and social variables is also analyzed.

Point 2. Line 17  Clarify “using tools to access ACEs.” Characterize the tools.

Response 2. Lines 18-19. Data were collected from a sample of 170 subjects, using tools to assess adverse childhood experiences, physical health-related quality of life, socioeconomic vulnerability, emotion regulation, coping strategies, attachment, and social support

Reviewer 2 Report

  • The introductory paragraph focuses on violence but the main variable of this study is ACE - adverse childhood experiences that are beyond just violence.  The authors may want to make it clear that their focuses are not only violence but adverse childhood.
  • Avoid using any causal words to refer to correlational findings. For example, in the abstract, the word "influence" (line 16) should be replaced with words that reflect correlations (e.g., associate, relate to). Other examples are "impact" (line 72), "direct effect/indirect influences" (line 98, line 234).
  • line 129-139 seems to be completely out of place. The earlier paragraphs were about the relationship between ACE and health and the "chains of risk", but then the authors suddenly mentioned the benefits of studying this in Spanish population without connecting the ideas logically. Also, it doesn't seem to make sense to say that "few studies have addressed it in... socially vulnerable population" - wouldn't people who experience ACE socially vulnerable by definition? 
  • Section 1.6 Aims can be better organized/represented if the authors clearly state the research questions or hypotheses, or draw a figure to represent the direct and indirect relationships they are examining. 
  • For the measurements, state the scale point. For example, for WHOQOL, the authors stated that there were five response options - what were the options? Also, provide at least one sample item for each measurement so that readers can get a better sense of how the concepts were measured. 
  • ForWHOQOL, did the authors only use the physical health domain? If so, please clearly state that. Also, how many items were the physical health domain? It seems unnecessary to mention the 26-item full scale when the authors only used the physical domain.  In addition, state how this domain was being scored (averaged of all items or sum)?
  • please also make sure to state scoring method for all measurements. 
  • The data analysis section should state clearly how each test corresponds to the hypotheses or research questions in section 1.6. 
  • What kind of stepwise method was used (backward, forward, mix)? Please state it when it is first mentioned in line 322 and in the results section when the test was first mentioned (line 375).
  • In the descriptive statistics section, the authors should report SD alongside the mean.
  • The section on correlation is very disorganized. Please use one paragraph to discuss the correlation pertaining to the validity of the questionnaire. And in fact, the discussion of validity should be reported in the measure section, not in the results. Results are tests relating to your hypotheses/research questions. 
  • again, please remove all causal language when reporting the results - this is a very serious mistake to misrepresent correlational findings as causal. 
  • Past tense should be used when describing past studies. 
  • a paragraph should contain at least three sentences. 

Author Response

Thanks
